# Spatial Distribution and Dietary Risk Assessment of Aflatoxins in Raw Milk and Dairy Feedstuff Samples from Different Climate Zones in China

**DOI:** 10.3390/toxins17010041

**Published:** 2025-01-16

**Authors:** Xueli Yang, Bolin Liu, Lei Zhang, Xiaodan Wang, Jian Xie, Jiang Liang

**Affiliations:** 1China National Center for Food Safety Risk Assessment, Beijing 100022, China; xueli_yang2024@163.com (X.Y.); zhanglei@cfsa.net.cn (L.Z.); wangxiaodan@cfsa.net.cn (X.W.); 2Xinjiang Uygur Autonomous Region Center for Disease Control and Prevention, No. 380, Jianquan 1st Street, Tianshan District, Urumqi 830001, China; 3Anhui Provincial Center for Disease Control and Prevention, No. 12560, Fuhua Avenue, Economic and Technological Development Zone, Hefei 230601, China; liubolin087@163.com (B.L.); xja@ahcdc.com.cn (J.X.)

**Keywords:** aflatoxin B1, aflatoxin M1, raw milk, feedstuff, spatial distribution, margin of exposure, cancer risk

## Abstract

This study aimed to explore the contamination of aflatoxins by investigating the spatial distribution of aflatoxin B1 (AFB1) in cow feedstuff and aflatoxin M1 (AFM1) in raw milk, and the potential health risks of AFM1 in milk and dairy products. Feedstuff and raw milk were collected from 160 pastures in three climate zones of China from October to November 2020. The results indicated the level of AFB1 and AFM1 ranged from 51.1 to 74.1 ng/kg and 3.0 to 7.0 ng/kg, respectively. Spatial analysis indicated the contamination was mostly concentrated in the temperate monsoon climate zone. On average, the estimated dietary exposure to AFM1 from milk and dairy products for Chinese consumers ranged from 0.0138 to 0.0281 ng/kg bw/day, with the MOE values below 10,000, and liver cancer risk of 0.00004–0.00009 cases/100,000 persons/year. For different groups, the average exposure to AFM1 was highest in the temperate monsoon climate zone and for toddlers.

## 1. Introduction

Milk is one of the most essential foods for humans, so government regulatory authorities pay great attention to quality control and risk assessment to prevent possible contamination. Aflatoxin M1 (AFM1), a hydroxylated metabolite that dairy animals produce in their milk after eating feedstuff contaminated with aflatoxin B1 (AFB1), is identified as the most significant toxin found in milk and dairy products [1,2]. There are 20 known types of aflatoxins reported, with aflatoxin B1 (AFB1), B2 (AFB2), G1 (AFG1), and G2 (AFG2) being the most prevalent [3]. Given their hepatotoxicity, carcinogenicity, and genotoxicity to human and animal, aflatoxins are classified as group 1 human carcinogens by the International Agency for Research on Cancer (IARC) [4]. Specifically, aflatoxin is a major risk factor for hepatocellular carcinoma (HCC), especially in the context of hepatitis B virus (HBV)-positive infection [5]. Although the carcinogenic potential of AFM1 is only one-tenth that of AFB1, long-term human exposure to AFM1 still raises concerns regarding the potential adverse health effects, particularly in children [6].

Due to its heat-resistance, AFM1 in milk is almost not destroyed by pasteurization or high-temperature sterilization [7], and can be detected in many dairy products, including milk powder, yogurt, cheese, butter, and so on [8]. Given the similar carcinogenic and immunosuppressive effects of AFM1 to AFB1 in humans and other animals [4,6], the maximum residue level (MRL) for AFM1 in milk and dairy products is strictly regulated in many countries and organizations. The Codex Alimentarius Commission (CAC) legal limit for AFM1 in milk, butter, and cheese are 0.5, 0.05, and 0.25 μg/kg, respectively [9]. The MRL set by the European Commission (EC) is 0.05 μg/kg in milk [10], which is 10-fold lower than the level in milk set by the CAC and China. The current China National Food Safety Standard (GB 2761-2017) stipulates the MRL as 0.5 μg/kg for AFM1 in milk and dairy products as well as in special dietary food products [11]. As the only mycotoxin regulated in raw milk, heat-treated milk, and milk for the manufacture of dairy products, the prevalence of AFM1 in milk worldwide was still reported to be 79.1% from global studies [12]. Published studies indicated that milk products with AFM1 contamination that markedly exceeded 0.5 μg/kg were found in Africa, Asia, Europe, and South America [13]. Recent studies in China have reported aflatoxin contamination in the cow feedstuff and dairy products from the market. Han et al. collected 200 milk samples from 10 major milk-producing provinces in China and revealed that 32.5% of the samples were positive for AFM1, with all of the levels below the limit of 0.5 μg/kg in China in 2010 [14]. Moreover, AFM1 was also reported to be 73.6% in UHT and pasteurized milk samples from central China, with average contamination values of 100.0 ng/L [15]. According to the report “2020 Statistical communique on National Economic and Social Development” released by China’s National Bureau of Statistics, milk production in China has reached 34.4 million tons, which increased by 7.5% as compared with 2019 [16]. As the consumption of milk has consistently increased over recent years, AFM1 residues in milk and dairy products have received increasing attention.

Since AFM1 is a hydroxylated derivative of AFB1, the concentration of AFM1 in milk was directly associated with the level of AFB1 contamination in dairy cow feedstuff [17]. The legal limits of AFB1 in dairy cow feedstuff varies by various authorities. The maximum allowable level of AFB1 for total aflatoxins (AFB1, AFB2, AFG1, and AFG2) in dairy cow feed has been set at 20 μg/kg in United States [18]. The EC has set a legal limit of 5 and 20 μg/kg for AFB1 in dairy cow feedstuff and in feed material, respectively [19]. The General Administration of Quality Supervision, Inspection and Quarantine of the People’s Republic of China and The Standardization Administration of the People’s Republic of China (GAQSIQ) have set the legal limit to 30 μg/kg in feedstuffs of vegetable origin [20]. It is known that the proliferation of aflatoxin-producing *Aspergillus* spp. is enhanced by warm temperatures, high humidity, and suboptimal storage conditions [21,22]. China’s milk production regions are mainly distributed in the temperate monsoon climate zone (40–47 degrees north latitude), including Neimenggu, Xinjiang, Hebei, and the northeast zones, with concentrations of 70% of the dairy cows and more than 60% of the raw milk. In an early study, 443 feedstuffs collected from 13 provinces of China from 2013 to 2015 were reported to be AFB1-positive, at 80.8% [23]. In another study in China, 35.1% of feedstuff samples were positive for AFB1, with 2.3% of samples exceeding the legal limit of 30 μg/kg in China [15]. An investigation on the occurrence of AFB1 in total mixed rations (TMRs) and AFM1 in raw and commercial milk from northern China revealed a linear relationship between AFB1 in TMR and AFM1 in raw milk [24]. Due to China’s vast territory, the aflatoxin contamination in feedstuff and raw milk might be affected by the diverse climate conditions. However, there is still a lack of current national comprehensive and systematic monitoring data on aflatoxin contamination in raw milk and feedstuff from the main production zones of China, which are characterized by their diverse geographical and climatic features. It is imperative to investigate the regional characteristics of AFM1 contamination in raw milk and feedstuff from pasture areas for precise prevention and control.

In the present study, the spatial contamination characteristics of AFB1 and AFM1 in feedstuff and raw milk from major pastures across the different climate zones of China were firstly investigated to reveal the regional risk profile of AFM1 exposure from milk and dairy products with the national consumption data of different age groups. The results can provide a factual basis for targeted regional risk prevention and control measures for AFM1 contamination in milk and dairy products.

## 2. Results

### 2.1. Occurrence of AFB1 in Feedstuff and AFM1 in Raw Milk

In the total of 160 feedstuff samples, 33 samples (20.6%) were positive for AFB1, with a mean value of 51.1–74.1 ng/kg and a maximum value of 1190 ng/kg. The values were all below the allowable limit of AFB1 in feedstuffs of vegetable origin (30 μg/kg) established by China [20]. No significant difference was observed in AFB1 among the three climate zones (*p* = 0.53), despite a relatively higher contamination level of 61.2–82.4 ng/kg in the temperate continental zone.

In the raw milk, 23.1% samples were detected with AFM1, with the mean level of 3.0–7.0 ng/kg and the maximum value of 43.6 ng/kg, which were all far below the China limit of 0.5 µg/kg [11] and the EC limit of 0.05 µg/kg for AFM1 in milk [10]. Furthermore, no statistically significant difference for AFM1 in raw milk was found between the three climate zones (*p* = 0.45). AFB1 and AFM1 contamination in feedstuffs and raw milk from the different climate zones in China are depicted in Table 1.

The spatial distribution of AFB1 in feedstuff and AFM1 in raw milk was analyzed using ArcGIS software (Figure 1). Among the 160 pastures, samples from 21 pastures (13.1%) were concurrently positive for feedstuff and raw milk (Figure 1a). The contamination of the feedstuff and raw milk samples was mostly concentrated in the temperate monsoon climate zone (*n* = 10). The feedstuff samples from Xinjiang, Ningxia, Shandong, Henan, and Jiangsu had relatively high concentrations of AFB1, with the mean levels ranging from 109.1 to 238.0 ng/kg (Figure 1b). The raw milk samples from Henan and Shandong were found to have comparatively high AFM1 contamination levels, with the mean vales ranging from 9.1 to 15.9 ng/kg (Figure 1c). In addition, the spatial correlation for AFB1 contamination in feedstuff and AFM1 contamination in raw milk across different provinces in China was not statistically significant (*p* > 0.05).

The correlation of aflatoxin contamination in feedstuff and raw milk was analyzed when aflatoxins were detected concurrently at 21 sampling points (Appendix A). In general, with the exception of a few samples from Beijing, Gansu, Henan, Ningxia, and Shaanxi, the contamination distribution of AFB1 in feedstuff and AFM1 in raw milk exhibited a similar trend within the same sampling sites. Moreover, Spearman’s non-parametric analysis revealed a significant positive correlation between AFB1 in feedstuff and AFM1 in raw milk at the same sampling sites (rs = 0.545, *p* < 0.001).

### 2.2. Consumption Levels of Milk and Dairy Products

The consumption of milk and dairy products in different climate zones was analyzed (Table 2). The daily consumption levels for milk and dairy products of the consumer population ranged from 98.74 to 192.02 g. A significant difference was observed in the consumption level for different climate zones (*p* < 0.05). For different climate zones, the average consumption level of milk and dairy products for consumers from the temperate continental zone was the highest (143.03 g/d), followed by the temperate monsoon zone (138.39 g/d) and subtropical monsoon zone (136.28 g/d). The daily consumption levels of milk and dairy products for the consumer population from different regions of China was also analyzed (Appendix A). Shanghai, which belongs to the subtropical monsoon climate zone, had the highest (192.02 g/d) consumption of milk and dairy products among the 23 provinces, followed by Guangxi (184.09 g/d). Furthermore, significant differences were found among different age groups. Consumers under the age of 18 years old had a significantly higher average consumption level of milk and dairy products as compared to adults (126.09 g/d) and the elderly (136.42 g/d). Among them, children were the highest (154.90 g/d), followed by adolescents (152.70 g/d) and toddlers (151.71 g/d).

### 2.3. AFM1 Dietary Exposure Assessment

Based on the contamination data of AFM1 in raw milk and the consumption data from the China Food Consumption Survey of 2017–2020, the daily exposure to AFM1 for the consumers of different age groups from different climate zones of China was estimated (Table 3). The AFM1 exposure from milk and dairy products was 0.0138–0.0281 ng/kg bw/day at the average consumption level, and 0.0482–0.0871 ng/kg bw/day at the high percentile (P95) consumption level. For the consumers from different climate zones at the average consumption level, the average exposure to AFM1 for consumers from the temperate monsoon zone was the highest (0.0152–0.0291 ng/kg bw/day), followed by the subtropical monsoon zone (0.0147–0.0290 ng/kg bw/day). At the high percentile (P95) consumption level, the exposure level for consumers from the temperate monsoon zone was 0.0618–0.0951 ng/kg bw/day, which was significantly higher than in the temperate continental zone. Among different age groups, toddlers were found with the highest mean exposure to AFM1 (0.0367–0.0730 ng/kg bw/day), followed by children (0.0184–0.0375 ng/kg bw/day). The high percentile (P95) of AFM1 exposure for toddlers was also the highest compared with the others, at 0.1157–0.2017 ng/kg bw/day.

The contribution of different milk and dairy products to the average daily exposure to AFM1 for the consumer population were ranked as follows: milk > yogurt > milk powder > cheese > cream > other dairy products (Figure 2). Milk, yogurt, and milk powder were the main contributors to the overall AFM1 average exposure for consumers, accounting for 66.31%, 20.21%, and 13.10%, respectively. Other dairy products, cream, and cheese all contributed less than 0.5% to the average daily exposure to AFM1 for the consumer population (Figure 2d). Due to the high milk consumption level in the temperate continental climate zone, the contribution rate of milk to AFM1 exposure among consumers (80.42%) was significantly higher than that in the temperate monsoon climate (68.93%) and subtropical monsoon climate (63.29%) (Figure 2a–c). In addition, the contribution rate of yogurt to the average daily exposure to AFM1 for the temperate monsoon climate (23.44%) was obviously higher as compared to the other climate zones (12.52% and 19.74%), while milk powder in the subtropical monsoon climate (16.66%) was higher than that in the temperate continental climate (5.83%) and temperate monsoon climate (7.32%).

Among the five age groups, the contribution rates of milk and yogurt ranged from 60.90% (toddlers) to 68.58% (children), and 12.96% (elderly) to 21.84% (children), respectively (Appendix A). Due to the relatively high milk consumption among toddlers and the elderly, the contribution rate of milk powder to AFM1 exposure was higher for toddlers (19.17%) and the elderly (19.90%) compared to children (9.35%), adolescents (12.19%), and adults (10.97%).

### 2.4. Risk Characterization/Cancer Risk Attributable to AFM1

Based on the MOE approach recommended by the EFSA, the MOE values for UB exposure scenarios to AFM1 from milk and dairy products were compared between the consumers of different age groups from the different climate zones (Figure 3). The results showed that the MOE values for the mean and P95 AFM1 exposure from milk and dairy products were far higher than 10,000 for the total consumers of China, which indicated that exposure to AFM1 from milk and dairy products was a low health concern for consumers of different age groups as well as from different regions.

According to the quantitative liver cancer risk approach proposed by the JECFA, the extra risk of HCC by AFM1 exposure from milk and dairy products was estimated (Table 4). The HBsAg^+^ prevalence rate in China was reported to be 7.18%, derived from the national seroepidemiology survey of China in 2006 [25]. Thus, the annual incidence of liver cancer caused by the exposure to 1 ng of AFM1 per kilogram of body weight for the Chinese population would be 0.003 cases/100,000 persons/year (average potency). Hence, for the consumers from the three climate zones, the additional cancer risk attributed to the mean exposure to AFM1 was 0.00001–0.00009 cases/100,000 persons/year at the average milk and dairy products consumption levels, and 0.00003–0.00029 cases/100,000 persons/year at the P95 percentile. For different age groups, the estimated liver cancer cases was 0.00002–0.00022 cases/100,000 persons/year at average consumption level, and 0.00007–0.00062 cases/100,000 persons/year at P95 percentile. Among them, toddlers were revealed to have the highest liver cancer risk, at 0.00011–0.00022 cases/100,000 persons/year for the mean, and 0.00211–0.00292 cases/100,000 persons/year for the P95 high consumption level.

As published in the 2018 China Tumor Registration Annual Report, the annual incidence of liver cancer in China in 2014 was 18.0 cases/100,000 persons/year [26]. The potential risk attributed to AFM1 exposure from milk and dairy products accounted on average to 0.0002–0.0005% of the overall annual liver cancer incidence, with 0.00003–0.0005% for the different climates groups and 0.0001–0.0012% for the different age groups. The highest potential liver cancer risk was estimated to be in toddlers, with the mean and P95 contribution rates of 0.0006–0.0012% and 0.0020–0.0035%, respectively (Table 4).

## 3. Discussion

The contamination level of AFB1 in feedstuff and AFM1 in raw milk were not significantly different in the three climate zones. In our study, the samples were mostly from well-known dairy enterprises of different sizes in China, with scientific and standardized feedstuff planting and storage technology. In general, the contamination level of AFB1 in feedstuff may be related with factors including geographical conditions, local temperature and humidity, the sampling season, feedstuff ingredients, storage methods, and farm management methods [27]. For well-known pastures, the feedstuff distribution, ratio, and supply are generally standardized. Such a unified and standardized management might result in the unification of feedstuff sources in different zones, and thus the differences in aflatoxin contamination levels in the feedstuff and raw milk in the different climate zones were not obvious, which was in line with our study. Although the pastures distribute feedstuff uniformly, the storage conditions and duration are critical to the control of AFB1 in feedstuff [28]. In our present study, the highest contamination level of AFB1 in feedstuff was found in the pasture located in the temperate continental zone, especially for Xinjiang and Ningxia provinces. The sampling was performed during October and November 2020, the early winter in Xinjiang and Ningxia provinces. According to the National Qinghai-Tibet Plateau Scientific Data Center [29], the mean temperatures for the period in Xinjiang and Ningxia provinces were 4.6 and 5.4 °C, respectively. When compared to zones with temperate monsoon and the subtropical monsoon climates, there has been an increase in the stored concentrated feedstuffs with maize as the main ingredient in feedstuff [30]. According to the previous studies, maize was particularly susceptible to contamination by AFB1 [15,31,32], which could explain the elevated levels of AFB1 contamination observed in zones with a temperate continental climate. Additionally, the concentration of AFM1 in raw milk during winter was significantly higher than in summer [33,34]. This increase may be attributed to the fact that feeds, such as silage and maize, are more prone to mildew during storage in winter, which can lead to spoilage and contribute to the AFB1 contamination in feedstuff [24]. It is worth noting that the AFM1-positive samples of raw milk were predominantly concentrated in the temperate monsoon climate zone, especially in the provinces of Henan and Shandong. Consistent with our results, Li et al. [33] investigated AFM1 contamination across major milk-producing areas of China over the course of four seasons, and found that the highest level of AFM1 contamination in raw milk were observed in Henan.

In our present study, the detection rate of AFM1 in raw milk (23.1%) was slightly higher than that in feedstuff (20.6%). The discrepancy in aflatoxin occurrence between feedstuff and raw milk could be attributed to the variation in the conversion rate and metabolic processes of AFB1 into AFM1 among different cows [35]. The carry-over rate from AFB1 to AFM1 was reported to range from 0.22% to 2.74%. This variation depends on various factors, including the dairy cow milk yield, stage of lactation, cow species, and the health of mammary alveolus cell membranes [36,37]. In addition, many published studies have investigated the timing of the AFB1 metabolism into AFM1 in dairy cows. Kuilman et al. [38] discovered that AFM1 formed within the first 2 to 8 h of incubating AFB1 in bovine hepatocytes. Frobish et al. [39] found that, after feeding cows with AFB1-contaminated feedstuff for 24 h, AFM1 could reach a stable level in milk, and could be eliminated by stopping feeding with AFB1-contaminated feedstuff for 3–4 days. Additionally, there may be batch-to-batch variation during the sampling process, where the feedstuff collected was inconsistent with the feedstuff consumed by the cow prior to milk production.

Studies in many countries around the world have reported the existence of AFB1 contamination in feedstuff [24,36,40], and cows will metabolize AFM1 after eating the feedstuff contaminated with AFB1 [2,41]. In accordance with our results, Spearman’s correlation analysis showed that the contamination of AFB1 in feedstuff was positively correlated with the AFM1 concentration in raw milk. Therefore, to reduce AFM1 contamination in raw milk, a standardized management program is required, including the appropriate frequency of irrigation, the harvest time, reducing the moisture in feedstuff (≤12%), and a dry and ventilated storage environment [21,22]. In addition, efficient bio-detoxifying methods should be developed to remove aflatoxins [42].

By reviewing studies on the prevalence of AFM1 in raw milk worldwide from 2010 to 2020 (Table 5), our analysis revealed that 37 out of 160 samples (23.1%) of raw milk were contaminated with AFM1, which was similar to the 30.1% observed in Turkey [43] and the 24.2% found in Croatia [44], yet higher than the 4.7% reported in the six provinces of China [33]. The present study found that the mean level of AFM1 contamination in raw milk was 3.0–7.0 ng/kg. This was lower than the levels reported in previous studies on AFM1 contamination in raw milk in China, with the values of 37.4 ± 18.7 [45] and 36.8 ± 43.6 ng/kg in 2016 [33], 15.9± 7.1 ng/kg in 2019 [46], and 110 ng/kg in 2020 [24]. Differences in AFM1 contamination levels in raw milk might be attributed to various factors, including climates, geographical environments, seasonal variations, and feedstuff systems [24]. Although the AFM1 levels observed in our study were significantly lower than those reported in previous reports, the contamination of AFM1 in the raw milk still should be limited to the lowest possible level to minimize health risks, based on the principle of “As Lowest as Reasonable Acceptable” (ALARA) [18] for genotoxic and carcinogenic substances proposed by the Food and Agriculture Organization (FAO).

The mean exposure to AFM1 from milk and dairy products for Chinese consumer populations ranged from 0.0138 to 0.0281 ng/kg bw/day. This was lower than the AFM1 exposure of 0.2 ng/kg bw/d estimated by JECFA’s report [52], in which the intake of AFM1 through milk in five geographical zones (Africa, Middle East, Latin America, Europe, and the Far East) were assessed at the 56th meeting in 2001, based on the global environmental monitoring system/Food (GEMS/Food) [53]. The exposure to AFM1 in our study was also significantly lower as compared to those in some European countries. For instance, in Italy, in the case of large-portion-size consumers, the exposure levels varied between 0.35 and 1.16 ng/kg bw/d [54]. Additionally, in Croatia, the exposure to the mean concentration of positive AFM1 milk samples during a five-year period ranged from 0.10 to 1.15 ng/kg bw/d [55]. The present assessment indicated that the extra liver cancer risk from AFM1-contaminated milk and dairy products among Chinese consumers contributed to only 0.0002–0.0005% to the current liver cancer incidence in China (18.0 cases/100,000 persons/year) [26]. Overall, the findings in this study suggested that the occurrence of AFM1 in milk and dairy products did not pose a significant public health risk in China. However, the risk of AFM1 exposure for children, particularly toddlers, should still be taken for concern due to their relatively higher milk consumption level and lower body weight, and the underdeveloped excretion organs [56].

There are still some uncertainties in the present study. Firstly, although the consumption data included dairy products, specific AFM1 contamination levels for the dairy products were not directly obtained. A certain coefficient was multiplied by the AFM1 contamination in raw milk in the present study without considering the AFM1 loss during processing, which could have potentially led to an underestimation of the exposure levels. Secondly, the sampling period of our study was in winter, and, while not covering all seasons, previous studies have shown that aflatoxin contamination in feedstuff and raw milk was significantly higher in winter than in other seasons [14,33]. Thus, winter sampling provides a more conservative estimate for aflatoxin exposure assessment. In addition, although this assessment collected raw milk samples from 160 representative pastures in China, covering major milk and dairy brands in the Chinese market, the sampling sites did not cover all of the zones in China, which may have led to a certain uncertainty in the exposure risk estimation.

## 4. Conclusions

In summary, the occurrence of AFB1 and AFM1 in feedstuff and raw milk samples from major pastures of China in our present investigation were all within the legal limits set by China and the EC. Although all of the samples containing AFB1 and AFM1 were well below the legal limits set by China for feedstuff and milk, the spatial analysis indicated that the contamination of feedstuff and raw milk samples was mostly concentrated in the temperate monsoon climate zone, with a correlation observed between the concentrations of AFB1 and AFM1. Long-term chronic exposure of AFM1 from milk and dairy product consumption were estimated to pose a low liver cancer risk to Chinese consumers. For different groups, the average exposure to AFM1 was the highest among consumers in the temperate monsoon climate zone and for toddlers. Further continuous AFM1 monitoring in milk and dairy products are essential for public health.

## 5. Materials and Methods

### 5.1. Sample Collection

#### 5.1.1. Geographical Distribution of Sampling Sites

Based on data from the National Bureau of Statistics (NBS) report on dairy industry information in China for 2017 [57], 160 sampling pastures were selected from 13,591 pastures by a simple random sampling method from late October to early November 2020 (three weeks), and paired feedstuff (*n* = 160) and raw milk (*n* = 160) were collected from dairy farms located in 25 provinces and cities in China (Figure 4). Based on the climatic characteristics of China, the sampling sites were classified as follows: temperate monsoon climate (*n* = 78), subtropical monsoon climate (*n* = 51), and temperate continental climate (*n* = 31). The monthly climate data were acquired from the National Qinghai-Tibet Plateau Scientific Data Center [29,58] (Table 6).

#### 5.1.2. Raw Milk and Feedstuff Samples Collection and Pretreatment

At each dairy farm, a pooled milk sample was collected directly from the milk holding tank to a new polypropylene (PP) bottle, and the paired feedstuffs were collected simultaneously. All of the samples were frozen and stored, and then delivered to the laboratory within 24 h.

The raw milk samples were directly collected from three primary milk storage tanks at the milk stations in the dairy farms. After stirring the milk holding tanks, 100 mL of the milk sample was collected and mixed to a total of 300 mL. The mixed raw milk samples were divided into five 50 mL centrifuge tubes, each containing about 40 mL of the raw milk samples, and stored at −20 °C until the analysis was performed. If there were only 1 or 2 primary milk storage tanks in the pasture, 300 mL was collected from 1 milk storage tank or 150 mL from 2 milk storage tanks.

The main ingredients for the feedstuff samples in the farms were identified as follows: maize silage, concentrate supplement for lactating cows, soybean meal, alfalfa, corn compression tablets, cottonseed, oat, beet pulp, corn meal and molasses, baking soda, water, and other minerals and vitamins. The multi-point sampling method was adopted in each pasture, and about 200 g of feedstuff sample was collected from at least 5 sampling sites. After full and uniform mixing, the feedstuff samples were divided into five 50 mL centrifuge tubes, sealed with tight caps and sealing films, and immediately stored at −20 °C until being analyzed.

#### 5.1.3. Sample Analyses

The China National Food Safety Standard GB 5009.22-2016 (first method) and GB 5009.24-2016 (first method) were used for analyzing the AFB1 in feedstuff and AFM1 in raw milk, respectively. The linear calibration curve of AFB1 was obtained with concentrations ranging from 0.10 to 50 ng/mL, and the coefficient of correlation was 0.9996 (r^2^). For AFM1, the linear range was 0.05 to 10.0 ng/mL, with a linearity of r^2^ = 0.9994. The limit of detection (LOD) and limit of quantitation (LOQ) were estimated using the signal-to-noise (S/N) ratios observed in the sample (feedstuff or raw milk) extracts of the less intensive mass transition using MassLynx (software version: 4.1, Waters, Milford, MA, USA). In general, the noise was determined from the baseline at a time interval of 0.2 min before the respective analyte peak in the sample spiked before the extraction. The LOD and LOQ values were calculated with S/N ratios of 3:1 and 10:1, respectively, for the spiked samples (*n* = 6) [59]. The LOD and LOQ values in the feedstuff were 0.03 and 0.1 µg/kg, respectively. In the case of AFM1, the values were 0.005 and 0.017 µg/kg in the raw milk.

The chromatographic system consisted of an Acquity UPLC coupled to a tandem mass spectrometer (Xevo TQ, Waters, Milford, MA, USA). The UPLC was equipped with a quaternary solvent delivery system, degasser, automatic sampler, and column heater. Chromatographic separations were performed on a Waters UPLC BEH C_18_ analytical column (100 mm × 2.1 mm I.D., and 1.7 µm particle size), kept at a constant temperature of 40 °C. Mobile phases A and B were 5 mmol/L ammonium acetate/water with 0.1% formic acid and acetonitrile, respectively. The chromatographic method held the initial mobile phase composition (32% B) constant for 0.5 min. Then, the content of B was increased up to 45% at 4.2 min, followed by a linear gradient to 95% B at 5.0 min, and held constant for 0.5 min at 95% B. The flow rate used was 0.4 mL/min.

The mass spectrometer was operated in the positive electrospray ionization mode (ESI+). Two transitions per analyte were monitored in the multiple reaction monitoring (MRM) mode using the following operation parameters: capillary voltage: 3000 V; drying gas: 1000 L/h; gas temperature: 500 °C; ion source temperature: 150 °C. Waters MassLynx 4.1 software was used for the method development, data acquisition, and quantitative analysis. The UPLC-MS/MS analysis conditions are demonstrated in Appendix A.

Feedstuff samples (5 g ± 0.01 g) were weighed out in 50 mL centrifuge tubes and then the isotope standard solution of 13C17-AFB1 was added, followed by vortexing for 30 s. After the addition of 20.0 mL of acetonitrile/water (84:16, *v*/*v*), the centrifuge tubes were capped, shaken for 20 min, and centrifuged for 10 min at 10,000 rpm. An aliquot of 4.0 mL was then taken and diluted by the addition of 46.0 mL of a mixture of phosphate buffer solution (PBS) containing 0.1% tween-20. Afterwards, the diluted sample was purified and enriched by an AFB1 immunoaffinity column. The retained analytes were eluted with 2 × 1.0 mL methanol and subsequently dried by vacuum during 1 min. This eluate was then evaporated until near-dryness by a gentle nitrogen stream, with a water bath temperature of 50 °C and a N2 pressure of 20 psi. The samples were then redissolved with 1.0 mL of acetonitrile and 5 mmol/L ammonium acetate/water containing 0.1% formic acid (32:68, *v*:*v*). Afterwards, the sample was filtered on a polytetrafluoroethylene (PTFE) 0.22 μm syringe filter. Finally, the extract was placed into an autosampler vial and injected in the UPLC-MS/MS system.

Raw samples (2 g ± 0.01 g) were weighed out in 50 mL centrifuge tubes, and then the isotope standard solution of 13C17-AFM1 was added, followed by vortexing for 30 s. After the addition of 10.0 mL methanol, the centrifuge tubes were capped, shaken for 3 min, and centrifuged at 4 °C for 10 min at 10,000 rpm. An aliquot of 10.0 mL was then taken and diluted by the addition of 40.0 mL PBS (containing 0.1% tween-20). Afterwards, the diluted sample was purified and enriched by an AFM1 immunoaffinity column. The retained analytes were eluted with 2 × 2.0 mL methanol and subsequently dried by vacuum during 1 min. This eluate was then evaporated until near-dryness by a gentle nitrogen stream, with a water bath temperature of 50 °C and a N2 pressure of 20 psi. The samples were then redissolved with 1.0 mL of acetonitrile and 5 mmol/L ammonium acetate/water containing 0.1% formic acid (32:68, *v*:*v*). Afterwards, the sample was filtered on a polytetrafluoroethylene (PTFE) 0.22 μm syringe filter. Finally, the extract was placed into an autosampler vial and injected in the UPLC-MS/MS system.

### 5.2. Consumption Data of Milk and Dairy Products

The dietary consumption data were collected using 3-day 24-h dietary review method. The food consumption data for milk, milk powder, yogurt, cheese, cream, and other dairy products were from the China Food Consumption Survey of 2017–2020 conducted by the China national center for food safety risk assessment [60]. A total of 55,678 participants aged 3 years and above were obtained from 23 provinces, autonomous zones, and municipalities in China. Age groups were used during data analysis, which was estimated for five different groups as toddlers (>3 years to ≤6 years), children (>6 years to ≤12 years), adolescents (>12 years to ≤18 year), adults (>18 years to ≤60 years), and the elderly (>60 years).

### 5.3. Exposure Assessment

The exposure to AFM1 from milk and dairy products was calculated by multiplying the daily consumption of milk and dairy products by the corresponding mean concentration of AFM1, and then dividing by body weight of individual consumer [28]. The exposure to AFM1 is calculated according to the following Equation (1):(1)$$Estimated\;daily\;intake\;(EDI)=\frac{\sum_{i=1}^{n}\left(C_{i}\times {CF}_{i}\times F_{i}\right)}{{bw}_{i}\times 1000}$$
where *EDI* is the estimated daily intake of AFM1 for milk and dairy product consumers (ng/kg bw/day); C_i_ is the mean AFM1 concentration of food category i (ng/kg); CF_i_ is the coefficient of food category i. The AFM1 concentration in different dairy products, including milk powder, yogurt, cheese, and cream, was estimated by multiplying by a certain coefficient from raw milk to dairy products. Specifically, the conversion coefficients of milk powder and cheese were estimated to be 8 and 5, respectively [12,61]; for milk, yogurt, and cream, it was estimated as 1 [28]. F_i_ is the consumption amount of food category i (g/day); bw_i_ refers to individual body weight (kg).

### 5.4. Risk Characterization

#### 5.4.1. Margin of Exposure (MOE) Approach

The MOE approach was used for the risk characterization of genotoxic and carcinogenic substances [62]. The EFSA considers an MOE of 10,000 or more, based on the BMDL_10_ from laboratory animal studies, as a low risk from a public health perspective and a low priority for risk management, while an MOE value lower than 10,000 indicates a high public health concern and priority for risk management measures. Based on male rat data, the BMDL_10_ of 0.4 µg/kg bw/day for the induction of HCC by AFB1, a carcinogenic potency factor of 0.1 relative to AFB1, was used to derive the BMDL_10_ of AFM1 as 4.0 µg/kg bw per day [63]. The calculation of the MOE was performed as Equation (2):*MOE* = BMDL_10_/*EDI*(2)

#### 5.4.2. Quantitative Liver Cancer Risk Approach

Based on the toxicological and epidemiological studies of AFB1 and the synergistic hepato-carcinogenic effect of AFB1 and HBV infection, the JECFA derived the quantitative model to estimate the AFB1 carcinogenic potency [64]. The carcinogenic potency of 1 ng/kg bw/day of AFM1 for HBV in surface antigen-positive (HBsAg^+^) and HBV in surface antigen-negative (HBsAg^−^) individuals was estimated to be 0.03 and 0.001 cases/100,000 persons/year, respectively [52]. Considering the HBsAg^+^ prevalence rate (*P*) in the total population, the extra liver cancer incidence per year attributed to AFM1 exposure from milk and dairy products were evaluated according to Equations (3) and (4), as follows:*Acerage potency* = 0.03 × *P* + 0.001 × (1 − *P*)(3)*Cancer risk* = *EDI* × *Acerage potency*
(4)

### 5.5. Statistical Analysis

All of the undetected results were replaced by 0 and the LOD as the lower bound (LB) and upper bound (UB), respectively [65]. The concentrations were presented as the mean, median, P75, P90, P95, and max (LB-UB). All of the statistical analyses for aflatoxins were conducted using SPSS (version 22.0, SPSS, Inc., Chicago, IL, USA). When the data did not conform to a normal distribution, non-parametric tests (the Kruskal–Wallis H test and Mann–Whitney U test) were used to compare the distribution differences in AFB1 and AFM1 contamination, and dietary exposure in different climate zones. Additionally, the Spearman rank correlation analysis was used to assess the correlations between the aflatoxins in feedstuff and raw milk, with the significance level set at *p* < 0.05. The spatial distribution of the aflatoxins was conducted using ArcGIS software (ArcGIS10.6.0 for desktop; Esri, Redlands, CA, USA), and Geoda software was used to analyze the spatial autocorrelation of the aflatoxin contamination levels. The locations of the sampling sites were converted into their geographical coordinates, expressed as longitude and latitude, using Geocoder (open-source; lbsyun.baidu.com, accessed from late October to early November 2020.).

## Figures and Tables

**Figure 1 toxins-17-00041-f001:**
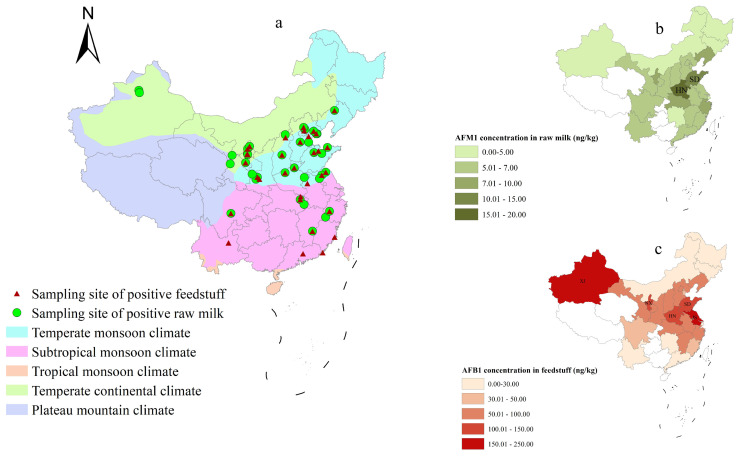
Spatial distribution of AFB1 in feedstuff and AFM1 in raw milk in different climate zones of China. (**a**) Spatial distribution of pastures with positive feedstuff and raw milk samples; (**b**) Spatial distribution of AFB1 contamination in feedstuff in different provinces of China; (**c**) Spatial distribution of AFM1 contamination in raw milk in different provinces of China. XJ: Xinjiang, NX: Ningxia, HN: Henan, SD: Shandong, and JS: Jiangsu.

**Figure 2 toxins-17-00041-f002:**
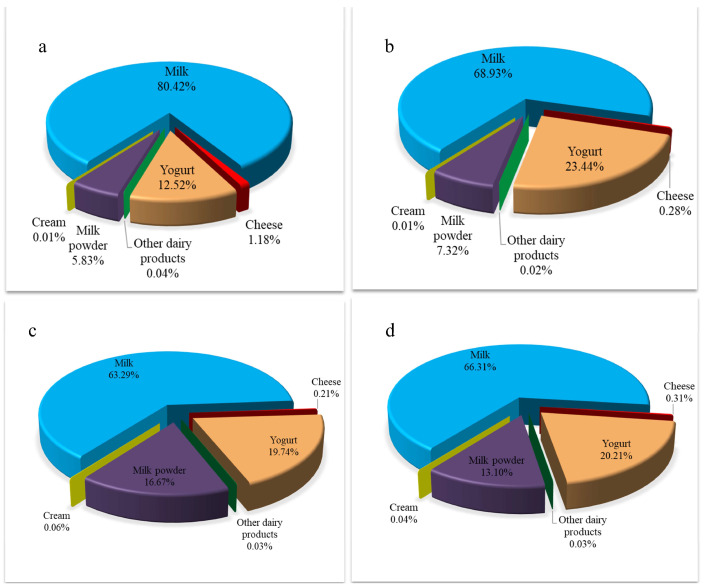
Contribution of different milk and dairy products to AFM1 daily exposure of consumers from different climate zones. (**a**) Temperate continental zone; (**b**) Temperate monsoon zone; (**c**) Subtropical monsoon zone; (**d**) Total consumer population.

**Figure 3 toxins-17-00041-f003:**
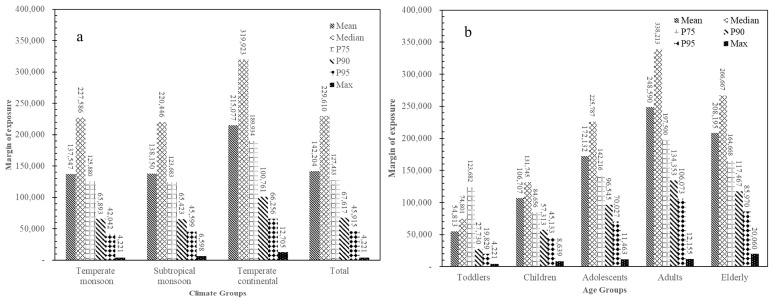
MOE values of the AFM1 exposure from milk and dairy products for consumers of different climate zones and age groups in China. (**a**) Climate groups; (**b**) Age groups. The values in the UB exposure scenario were used to calculate the MOE.

**Figure 4 toxins-17-00041-f004:**
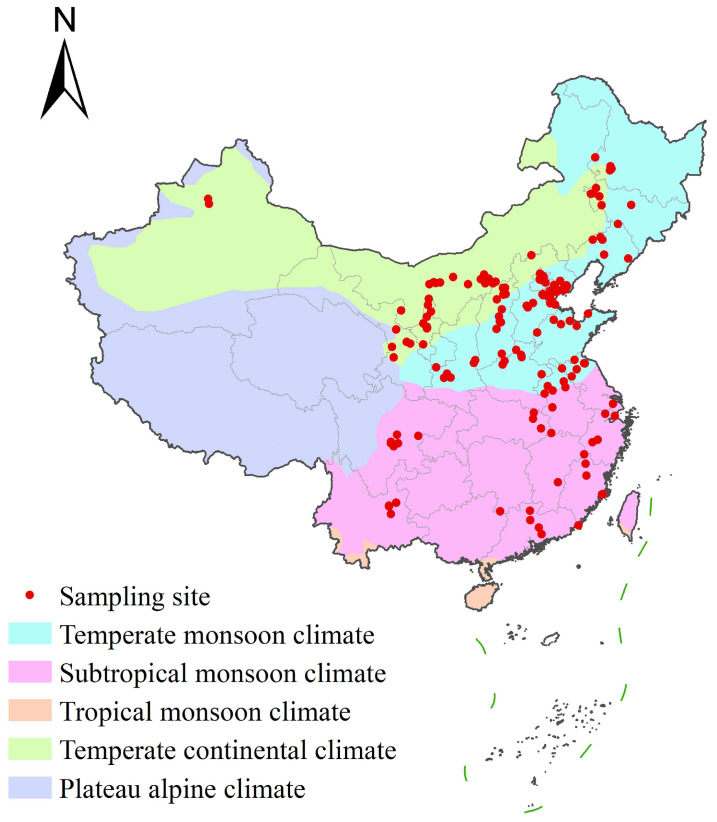
The spatial geographic distribution of the pasture sampling sites in China.

**Table 1 toxins-17-00041-t001:** AFB1 and AFM1 contamination in feedstuffs and raw milk from different climate zones in China.

Climate	AFB1								AFM1						
	Number of Samples	Positive Samples (%)	Mean (ng/kg)	Median (ng/kg)	P75 (ng/kg)	P90 (ng/kg)	P95 (ng/kg)	Max (ng/kg)	Positive Samples (%)	Mean (ng/kg)	Median (ng/kg)	P75 (ng/kg)	P90 (ng/kg)	P95 (ng/kg)	Max (ng/kg)
Temperate monsoon	78	14 (17.9)	43.7–67.9	15	30	139.6	343.3	560	15 (19.0)	3.3–7.4	2.5	5	13.8	25.9	43.6
Subtropical monsoon	51	14 (27.4)	56.2–78.6	15	30	157.4	368.8	1190	13 (25.5)	3.2–6.8	2.5	8.6	13.4	16.2	17.4
Temperate continental	31	5 (16.2)	61.2–82.4	15	75.7	228.2	345	492	9 (29.0)	1.9–6.1	2.5	5	10.3	16.8	16.9
Total	160	33 (20.6)	51.1–74.1	15	30	184.9	336.1	1190	37 (23.1)	3.0–7.0	2.5	5	13.3	16.7	43.6

**Table 2 toxins-17-00041-t002:** Consumption of milk and dairy products in different climate zones and age groups in China.

Population	Number of Consumers	Percentage of Milk and Dairy Product Consumers (%)	Mean Consumption (g/Day)	95th Percentile Consumption (g/Day)
Temperate monsoon	4831	31.27	138.39	300.00
Subtropical monsoon	9868	27.16	136.28	300.00
Temperate continental	1338	34.40	143.03	300.00
Toddlers	2146	63.89	151.71	340.00
Children	2643	55.24	154.90	330.00
Adolescents	1265	50.06	152.70	333.33
Adults	8258	22.47	126.09	250.00
Elderly	1725	20.88	136.42	300.00
Total	16,037	28.80	137.48	300.00

**Table 3 toxins-17-00041-t003:** AFM1 daily exposure of milk and dairy product consumers of different age groups from different climate zones of China.

Population	EDI (ng/kg bw/d)					
	Mean	Median	P75	P90	P95	Max
Temperate monsoon	0.0152–0.0291	0.0066–0.0176	0.0168–0.0318	0.0368–0.0607	0.0618–0.0951	0.6847–0.9476
Subtropical monsoon	0.0147–0.0290	0.0086–0.0181	0.0166–0.0323	0.0316–0.0611	0.0477–0.0877	0.2878–0.6062
Temperate continental	0.0017–0.0186	0.0000–0.0125	0.0017–0.0211	0.0049–0.0397	0.0093–0.0604	0.0327–0.3148
Toddlers	0.0367–0.0730	0.0236–0.0535	0.0166–0.0323	0.0879–0.1442	0.1157–0.2017	0.6847–0.9476
Children	0.0184–0.0375	0.0132–0.0304	0.0243–0.0473	0.0406–0.0698	0.0574–0.0886	0.3661–0.4630
Adolescents	0.0117–0.0232	0.0077–0.0177	0.0148–0.0281	0.0261–0.0414	0.0377–0.0571	0.2046–0.3490
Adults	0.0076–0.0161	0.0049–0.0118	0.0097–0.0203	0.0165–0.0298	0.0226–0.0377	0.2378–0.3291
Elderly	0.0091–0.0192	0.0057–0.0150	0.0119–0.0243	0.0203–0.0341	0.0289–0.0465	0.1115–0.1994
Total	0.0138–0.0281	0.0073–0.0174	0.0154–0.0314	0.0311–0.0592	0.0482–0.0871	0.6847–0.9476

**Table 4 toxins-17-00041-t004:** Liver cancer risk of AFM1 exposure from milk and dairy product consumer populations in China.

Population	Extra Liver Cancer Risk from the Average AFM1 Exposure (Cases/100,000 Persons/Year) *											
	Mean		Median		P75		P90		P95		Max	
	LB	UB	LB	UB	LB	UB	LB	UB	LB	UB	LB	UB
Temperate monsoon	0.00005	0.00009	0.00002	0.00005	0.00005	0.00010	0.00011	0.00019	0.00019	0.00029	0.00211	0.00292
Subtropical monsoon	0.00005	0.00009	0.00003	0.00006	0.00005	0.00010	0.00010	0.00019	0.00015	0.00027	0.00089	0.00187
Temperate continental	0.00001	0.00006	0.00000	0.00004	0.00001	0.00006	0.00002	0.00012	0.00003	0.00019	0.00010	0.00097
Toddlers	0.00011	0.00022	0.00007	0.00016	0.00005	0.00010	0.00027	0.00044	0.00036	0.00062	0.00211	0.00292
Children	0.00006	0.00012	0.00004	0.00009	0.00007	0.00015	0.00013	0.00021	0.00018	0.00027	0.00113	0.00143
Adolescents	0.00004	0.00007	0.00002	0.00005	0.00005	0.00009	0.00008	0.00013	0.00012	0.00018	0.00063	0.00107
Adults	0.00002	0.00005	0.00002	0.00004	0.00003	0.00006	0.00005	0.00009	0.00007	0.00012	0.00073	0.00101
Elderly	0.00003	0.00006	0.00002	0.00005	0.00004	0.00007	0.00006	0.00010	0.00009	0.00014	0.00034	0.00061
Total	0.00004	0.00009	0.00002	0.00005	0.00005	0.00010	0.00010	0.00018	0.00015	0.00027	0.00211	0.00292
	**Liver Cancer Risk Contribution of the Mean AFM1 Exposure (%)**											
Temperate monsoon	0.0003	0.0005	0.0001	0.0003	0.0003	0.0005	0.0006	0.0010	0.0011	0.0016	0.0117	0.0162
Subtropical monsoon	0.0003	0.0005	0.0001	0.0003	0.0003	0.0006	0.0005	0.0010	0.0008	0.0015	0.0049	0.0104
Temperate continental	0.0000	0.0003	0.0000	0.0002	0.0000	0.0004	0.0001	0.0007	0.0002	0.0010	0.0006	0.0054
Toddlers	0.0006	0.0012	0.0004	0.0009	0.0003	0.0006	0.0015	0.0025	0.0020	0.0035	0.0117	0.0162
Children	0.0003	0.0006	0.0002	0.0005	0.0004	0.0008	0.0007	0.0012	0.0010	0.0015	0.0063	0.0079
Adolescents	0.0002	0.0004	0.0001	0.0003	0.0003	0.0005	0.0004	0.0007	0.0006	0.0010	0.0035	0.0060
Adults	0.0001	0.0003	0.0001	0.0002	0.0002	0.0003	0.0003	0.0005	0.0004	0.0006	0.0041	0.0056
Elderly	0.0002	0.0003	0.0001	0.0003	0.0002	0.0004	0.0003	0.0006	0.0005	0.0008	0.0019	0.0034
Total	0.0002	0.0005	0.0001	0.0003	0.0003	0.0005	0.0005	0.0010	0.0008	0.0015	0.0117	0.0162

* “Heat map” (scale: green–yellow–red) of the liver cancer risk of AFM1 exposure for milk and dairy product consumers in China.

**Table 5 toxins-17-00041-t005:** AFM1 occurrence in raw milk in different countries from 2010 to 2020.

Continent	Country	Year of Sampling and Season/Month	Number of Raw Milk Samples	Number of Positive Samples (%)	Mean/Mean ± SD/Range	Exceeding EC Legal Limit (>50 ng/L) in Samples (%)
Europe	Serbia [47]	Four seasons of 2013–2014	678	540 (79.6)	0.282 ± 0.358 (µg/kg)	382 (56.3)
	Croatia [44]	July to September 2013	194	47 (24.2)	20.6 ± 18.8 (ng/L)	13 (6.7)
America	Brazil [48]	August 2009 to February 2010	129	129 (100)	0.0195 0.0021 (µg/L)	18 (14.0)
	Argentina [49]	September 2012 to August 2013	160	62 (38.8)	0.037 (µg/L)	12 (7.5)
Africa	Tanzania [50]	February 2014	37	31 (83.8)	<LOD—2.007 (ng/mL)	31 (83.8)
	Nigeria [51]	July 2020	77	76 (99)	92 (ng/L)	32 (42)
Asia	Turkey [43]	January 2012 and December 2012	176	53 (30.1)	0.153 (µg/kg)	30 (17)
	Pakistan [40]	March 2017 until February 2018	75	41 (54.7)	68.4 ± 7.4 (ng/L)	17 (22.7)
	China [45]	October 2016	136	85 (62.5)	37.4 ± 18.7 (ng/kg)	8 (5.9)
	China [33]	Four seasons of 2016	5650	267 (4.7)	36.8 ± 43.6 (ng/L)	63 (1.1)
	China [46]	November 01 2018 to March 31 2019	133	100 (75.2)	15.9 ± 7.1 (ng/L)	0 (0)
	China [24]	November 2019 to January 2020	84	- *	110 (ng/kg)	29 (34.5)
	China (this study)	Late October to early November 2020	160	37 (23.1)	3.0–7.0 (ng/kg)	0 (0)

* Not mentioned in the reference.

**Table 6 toxins-17-00041-t006:** Information for the sampling sites from the 25 provinces covering the different climate zones of China.

Climate Zones ^a^	Provinces	Milk Production (Ten Thousand Tons)	Climate ^b^		Number of Sampling Points in Different Pasture Sizes ^c^					Total
			Temperature (°C)	Precipitation (mm)	≤199	200–499 ^c^	500–999	1000~	Subtotal	
Temperate monsoon climate	Liaoning	120.7	4.9	287.3	2	1	/	3	6	78
	Jilin	34.4	3.0	133.4	/	3	1	2	6	
	Heilongjiang	468.4	1.3	95.9	1	1	/	2	4	
	Beijing	37.4	7.9	80.9	/	5	1	3	9	
	Tianjin	52.1	9.9	98.2	1	1	3	5	10	
	Hebei	388.3	8.7	145.8	2	3	3	10	18	
	Henan	212.9	11.8	322.2	/	1	2	3	6	
	Shandong	231.3	11.3	306.6	/	2	1	4	7	
	Shanxi	78.1	4.6	118.6	/	3	2	7	12	
Subtropical monsoon climate	Shaanxi	156.9	9.4	527.1	1	1	2	2	6	51
	Jiangsu	49.0	13.6	281.6	1	1	3	3	8	
	Anhui	29.8	13.7	490.9	1	1	2	3	7	
	Shanghai	36.4	17.0	226.5	/	1	/		1	
	Zhejiang	14.3	17.6	404.7	/	2	/	1	3	
	Jiangxi	9.5	18.1	460.5	/	1	/		1	
	Hubei	12.8	15.0	882.0	4	/	/		4	
	Hunan	6.1	18.1	460.5	/	/	/	1	1	
	Sichuan	63.8	15.6	603.2	1	/	2	3	6	
	Yunnan	64.5	13.0	278.9	1	1	2		4	
	Guangdong	13.9	21.8	140.3	/	1	1	2	4	
	Fujian	13.1	17.1	284.3	/	/	5	1	6	
	Guizhou	4.4	/	/	/	/	/	/	0	
	Chongqing	5.1	/	/	/	/	/	/	0	
	Guangxi	8.1	/	/	/	/	/	/	0	
Temperate continental climate	Neimenggu	599.6	2.4	76.9	1	2	4	6	13	31
	Xinjiang	200.3	4.6	44.5	/	1	1	/	2	
	Ningxia	160.1	5.4	91.3	/	2	2	5	9	
	Gansu	41.0	4.2	104.9	/	1	2	4	7	
Tropical monsoon climate	Hainan	0.5	/	/	/	/	/	/	0	0
Plateau alpine climate	Xizang	42.0	/	/	/	/	/	/	0	0
	Qinghai	33.2	/	/	/	/	/	/	0	

^(a)^ China’s climate zoning data were obtained from the Chinese Academy of Sciences’ Resource and Environmental Science Data Center. ^(b)^ Average temperature and precipitation in October and November 2020 at each sampling site. ^(c)^ Number of cows in each pasture.

## Data Availability

The original contributions presented in this study are included in the article and Appendix A. Further inquiries can be directed to the corresponding author.

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
