# Peer review of "Spatial Distribution and Dietary Risk Assessment of Aflatoxins in Raw Milk and Dairy Feedstuff Samples from Different Climate Zones in China"

_toxins, 2025, doi:10.3390/toxins17010041_

Round 1

Reviewer 1 Report

Comments and Suggestions for Authors

1) Authors are required to include all the analytical conditions in the manuscript, including the instrument model, temperature program for the mass spectrometer, MRM conditions, and elution program used.

2) The authors are required to provide a detailed description of the methodology utilized for calculating the Limit of Detection (LOD). And LOQ needs to be provided

3) The manuscript or supplementary data needs to include chromatograms of the bank solvent, blank sample, LOD level, and LOQ level. 

Reviewer 2 Report

Comments and Suggestions for Authors

Spatial distribution and dietary risk assessment of aflatoxins in raw milk and dairy feedstuff samples from different climate zones in China

The manuscript is reviewed for consideration in toxins. The topic is very interesting. The methodology is well-defined and results are reproducible. The tables and figures are well aligned. However, the following points should be considered

In tables standard deviation values were not given which is very necessary, please incorporate them.

Furthermore, the normality analysis and ANOVA analysis were not applied, which is also a key point in the validity of the results.

Conclusion

The conclusion should be composed with the main highlights of the results. 

Reviewer 3 Report

Comments and Suggestions for Authors

General Comment:

The article describes the study of contamination and spatial distribution of Aflatoxin B1 in Feedstuffs and Aflatoxin M1 in milk in different climate zones of China in two months period 2020. Using relevant software, the authors found significant positive correlation between AFB1 in Feedstuffs and AFM1 in raw milk. Exposure assessment has been made by evaluation of the contamination data (AFM1 in raw milk) and the consumption data in the investigated region. The authors found important differences among different age groups and different climate zones, showing the importance of this study. In addition, interesting results were achieved for different products (milk, yogurt, milk powder, cheese, cream and other dairy products) in respect to their consumption in different climate zones.

The article is interesting and well written. There are only several issues that need to be addressed before acceptance of the article for publishing in your Journal and these are listed below.

Specific Questions:

Page 1, Lines 25-26. „Great attention is paid to quality control and risk assessment“ by whom?

Page 1, Line 32. Please check spelling for „Group 1“

Page 1, Lines 34-36. Please check grammar and style.

Page 1, Lines 39-40. The authors should include short explanation about the statement „high potential for hepatocarcinogenesis“

Page 2, Lines 83-85. Please check grammar and style.

Page 5, Lines 112-113. Please check grammar and style.

Page 14, Lines 304-305; 31-318 and 332-339. Please check grammar and style.

Page 15, Line 346. How long is actually this period form „late October to early November 2020“, one week or? Please explain.

Page 18, Line 362. „All samples were frozen and stored.“ How did the authors evaluate the results in Feed in respect to moisture ratio?
